# Link Prediction Based on Graph Neural Networks

**Muhan Zhang**
Department of CSE
Washington University in St. Louis
muhan@wustl.edu

**Yixin Chen**
Department of CSE
Washington University in St. Louis
chen@cse.wustl.edu

## Abstract

Link prediction is a key problem for network-structured data. Link prediction heuristics use some score functions, such as common neighbors and Katz index, to measure the likelihood of links. They have obtained wide practical uses due to their simplicity, interpretability, and for some of them, scalability. However, every heuristic has a strong assumption on when two nodes are likely to link, which limits their effectiveness on networks where these assumptions fail. In this regard, a more reasonable way should be learning a suitable heuristic from a given network instead of using predefined ones. By extracting a local subgraph around each target link, we aim to learn a function mapping the subgraph patterns to link existence, thus automatically learning a "heuristic" that suits the current network. In this paper, we study this heuristic learning paradigm for link prediction. First, we develop a novel $\gamma$-decaying heuristic theory. The theory unifies a wide range of heuristics in a single framework, and proves that all these heuristics can be well approximated from local subgraphs. Our results show that local subgraphs reserve rich information related to link existence. Second, based on the $\gamma$-decaying theory, we propose a new method to learn heuristics from local subgraphs using a graph neural network (GNN). Its experimental results show unprecedented performance, working consistently well on a wide range of problems.

## 1  Introduction

Link prediction is to predict whether two nodes in a network are likely to have a link [1]. Given the ubiquitous existence of networks, it has many applications such as friend recommendation [2], movie recommendation [3], knowledge graph completion [4], and metabolic network reconstruction [5].

One class of simple yet effective approaches for link prediction is called heuristic methods. Heuristic methods compute some heuristic node similarity scores as the likelihood of links [1, 6]. Existing heuristics can be categorized based on the maximum hop of neighbors needed to calculate the score. For example, common neighbors (CN) and preferential attachment (PA) [7] are **first-order** heuristics, since they only involve the one-hop neighbors of two target nodes. Adamic-Adar (AA) and resource allocation (RA) [8] are **second-order** heuristics, as they are calculated from up to two-hop neighborhood of the target nodes. We define $h$-*order heuristics* to be those heuristics which require knowing up to $h$-hop neighborhood of the target nodes. There are also some **high-order** heuristics which require knowing the entire network. Examples include Katz, rooted PageRank (PR) [9], and SimRank (SR) [10]. Table 3 in Appendix A summarizes eight popular heuristics.

Although working well in practice, heuristic methods have strong assumptions on when links may exist. For example, the common neighbor heuristic assumes that two nodes are more likely to connect if they have many common neighbors. This assumption may be correct in social networks, but is shown to fail in protein-protein interaction (PPI) networks – two proteins sharing many common neighbors are actually less likely to interact [11].

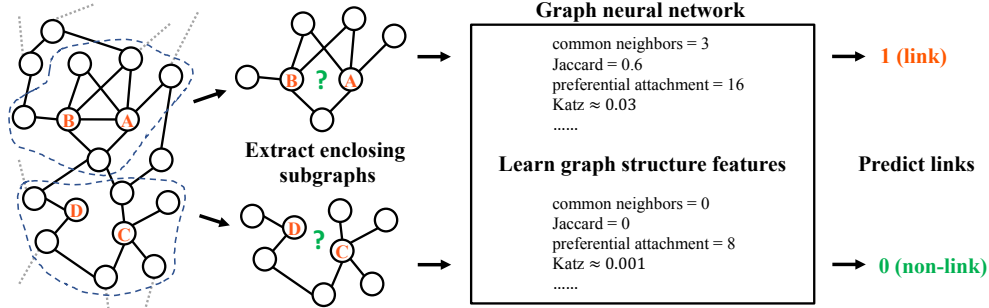

**Figure 1:** The SEAL framework. For each target link, SEAL extracts a local enclosing subgraph around it, and uses a GNN to learn general graph structure features for link prediction. Note that the heuristics listed inside the box are just for illustration – the learned features may be completely different from existing heuristics.

In fact, the heuristics belong to a more generic class, namely *graph structure features*. Graph structure features are those features located inside the observed node and edge structures of the network, which can be calculated directly from the graph. Since heuristics can be viewed as predefined graph structure features, a natural idea is to automatically learn such features from the network. Zhang and Chen [12] first studied this problem. They extract local enclosing subgraphs around links as the training data, and use a fully-connected neural network to learn which enclosing subgraphs correspond to link existence. Their method called Weisfeiler-Lehman Neural Machine (WLNM) has achieved state-of-the-art link prediction performance. The *enclosing subgraph* for a node pair $(x, y)$ is the subgraph induced from the network by the union of $x$ and $y$'s neighbors up to $h$ hops. Figure 1 illustrates the 1-hop enclosing subgraphs for $(A, B)$ and $(C, D)$. These enclosing subgraphs are very informative for link prediction – all first-order heuristics such as common neighbors can be directly calculated from the 1-hop enclosing subgraphs.

However, it is shown that high-order heuristics such as rooted PageRank and Katz often have much better performance than first and second-order ones [6]. To effectively learn good high-order features, it seems that we need a very large hop number $h$ so that the enclosing subgraph becomes the entire network. This results in unaffordable time and memory consumption for most practical networks. But do we really need such a large $h$ to learn high-order heuristics?

Fortunately, as our first contribution, we show that we do not necessarily need a very large $h$ to learn high-order graph structure features. We dive into the inherent mechanisms of link prediction heuristics, and find that most high-order heuristics can be unified by a *$\gamma$-decaying theory*. We prove that, under mild conditions, any $\gamma$-decaying heuristic can be effectively approximated from an $h$-hop enclosing subgraph, where the approximation error decreases at least exponentially with $h$. This means that we can safely use even a small $h$ to learn good high-order features. It also implies that the "effective order" of these high-order heuristics is not that high.

Based on our theoretical results, we propose a novel link prediction framework, SEAL, to learn general graph structure features from **local** enclosing subgraphs. SEAL fixes multiple drawbacks of WLNM. First, a graph neural network (GNN) [13, 14, 15, 16, 17] is used to replace the fully-connected neural network in WLNM, which enables better graph feature learning ability. Second, SEAL permits learning from not only subgraph structures, but also latent and explicit node features, thus absorbing multiple types of information. We empirically verified its much improved performance.

Our contributions are summarized as follows. 1) We present a new theory for learning link prediction heuristics, justifying learning from **local** subgraphs instead of entire networks. 2) We propose SEAL, a novel link prediction framework based on GNN (illustrated in Figure 1). SEAL outperforms all heuristic methods, latent feature methods, and recent network embedding methods by large margins. SEAL also outperforms the previous state-of-the-art method, WLNM.

## 2   Preliminaries

**Notations**   Let $G = (V, E)$ be an undirected graph, where $V$ is the set of vertices and $E \subseteq V \times V$ is the set of observed links. Its adjacency matrix is $A$, where $A_{i,j} = 1$ if $(i, j) \in E$ and $A_{i,j} = 0$

otherwise. For any nodes $x, y \in V$, let $\Gamma(x)$ be the 1-hop neighbors of $x$, and $d(x, y)$ be the shortest path distance between $x$ and $y$. A walk $w = \langle v_0, \cdots, v_k \rangle$ is a sequence of nodes with $(v_i, v_{i+1}) \in E$. We use $|\langle v_0, \cdots, v_k \rangle|$ to denote the length of the walk $w$, which is $k$ here.

**Latent features and explicit features** Besides graph structure features, latent features and explicit features are also studied for link prediction. **Latent feature methods** [3, 18, 19, 20] factorize some matrix representations of the network to learn a low-dimensional latent representation/embedding for each node. Examples include matrix factorization [3] and stochastic block model [18] etc. Recently, a number of network embedding techniques have been proposed, such as DeepWalk [19], LINE [21] and node2vec [20], which are also latent feature methods since they implicitly factorize some matrices too [22]. **Explicit features** are often available in the form of node attributes, describing all kinds of side information about individual nodes. It is shown that combining graph structure features with latent features and explicit features can improve the performance [23, 24].

**Graph neural networks** Graph neural network (GNN) is a new type of neural network for learning over graphs [13, 14, 15, 16, 25, 26]). Here, we only briefly introduce the components of a GNN since this paper is not about GNN innovations but is a novel application of GNN. A GNN usually consists of 1) *graph convolution layers* which extract local substructure features for individual nodes, and 2) a *graph aggregation layer* which aggregates node-level features into a graph-level feature vector. Many graph convolution layers can be unified into a message passing framework [27].

**Supervised heuristic learning** There are some previous attempts to learn supervised heuristics for link prediction. The closest work to ours is the Weisfeiler-Lehman Neural Machine (WLNM) [12], which also learns from local subgraphs. However, WLNM has several drawbacks. Firstly, WLNM trains a fully-connected neural network on the subgraphs' adjacency matrices. Since fully-connected neural networks only accept fixed-size tensors as input, WLNM requires truncating different subgraphs to the same size, which may lose much structural information. Secondly, due to the limitation of adjacency matrix representations, WLNM cannot learn from latent or explicit features. Thirdly, theoretical justifications are also missing. We include more discussion on WLNM in Appendix D. Another related line of research is to train a supervised learning model on different heuristics' combination. For example, the path ranking algorithm [28] trains logistic regression on different path types' probabilities to predict relations in knowledge graphs. Nickel et al. [23] propose to incorporate heuristic features into tensor factorization models. However, these models still rely on predefined heuristics – they cannot learn general graph structure features.

## 3 A theory for unifying link prediction heuristics

In this section, we aim to understand deeper the mechanisms behind various link prediction heuristics, and thus motivating the idea of learning heuristics from local subgraphs. Due to the large number of graph learning techniques, note that we are not concerned with the generalization error of a particular method, but focus on the information reserved in the subgraphs for calculating existing heuristics.

**Definition 1.** *(Enclosing subgraph) For a graph $G = (V, E)$, given two nodes $x, y \in V$, the $h$-hop enclosing subgraph for $(x, y)$ is the subgraph $G_{x,y}^h$ induced from $G$ by the set of nodes $\{ i \mid d(i, x) \leq h \text{ or } d(i, y) \leq h \}$.*

The enclosing subgraph describes the "$h$-hop surrounding environment" of $(x, y)$. Since $G_{x,y}^h$ contains all $h$-hop neighbors of $x$ and $y$, we naturally have the following theorem.

**Theorem 1.** *Any $h$-order heuristic for $(x, y)$ can be accurately calculated from $G_{x,y}^h$.*

For example, a 2-hop enclosing subgraph will contain all the information needed to calculate any first and second-order heuristics. However, although first and second-order heuristics are well covered by local enclosing subgraphs, an extremely large $h$ seems to be still needed for learning high-order heuristics. Surprisingly, our following analysis shows that **learning high-order heuristics is also feasible with a small** $h$. We support this first by defining the $\gamma$-decaying heuristic. We will show that under certain conditions, a $\gamma$-decaying heuristic can be very well approximated from the $h$-hop enclosing subgraph. Moreover, we will show that almost all well-known high-order heuristics can be unified into this $\gamma$-decaying heuristic framework.

**Definition 2.** *($\gamma$-decaying heuristic) A $\gamma$-decaying heuristic for $(x, y)$ has the following form:*

$$\mathcal{H}(x, y) = \eta \sum_{l=1}^{\infty} \gamma^l f(x, y, l), \tag{1}$$

*where $\gamma$ is a decaying factor between 0 and 1, $\eta$ is a positive constant or a positive function of $\gamma$ that is upper bounded by a constant, $f$ is a nonnegative function of $x, y, l$ under the the given network.*

Next, we will show that under certain conditions, a $\gamma$-decaying heuristic can be approximated from an $h$-hop enclosing subgraph, and the approximation error decreases at least exponentially with $h$.

**Theorem 2.** *Given a $\gamma$-decaying heuristic $\mathcal{H}(x, y) = \eta \sum_{l=1}^{\infty} \gamma^l f(x, y, l)$, if $f(x, y, l)$ satisfies:*

- *(property 1) $f(x, y, l) \leq \lambda^l$ where $\lambda < \frac{1}{\gamma}$; and*
- *(property 2) $f(x, y, l)$ is calculable from $G_{x,y}^h$ for $l = 1, 2, \cdots, g(h)$, where $g(h) = ah + b$ with $a, b \in \mathbb{N}$ and $a > 0$,*

*then $\mathcal{H}(x, y)$ can be approximated from $G_{x,y}^h$ and the approximation error decreases at least exponentially with $h$.*

*Proof.* We can approximate such a $\gamma$-decaying heuristic by summing over its first $g(h)$ terms.

$$\widetilde{\mathcal{H}}(x, y) := \eta \sum_{l=1}^{g(h)} \gamma^l f(x, y, l). \tag{2}$$

The approximation error can be bounded as follows.

$$|\mathcal{H}(x, y) - \widetilde{\mathcal{H}}(x, y)| = \eta \sum_{l=g(h)+1}^{\infty} \gamma^l f(x, y, l) \leq \eta \sum_{l=ah+b+1}^{\infty} \gamma^l \lambda^l = \eta(\gamma\lambda)^{ah+b+1}(1 - \gamma\lambda)^{-1}. \quad \square$$

In practice, a small $\gamma\lambda$ and a large $a$ lead to a faster decreasing speed. Next we will prove that three popular high-order heuristics: Katz, rooted PageRank and SimRank, are all $\gamma$-decaying heuristics which satisfy the properties in Theorem 2. First, we need the following lemma.

**Lemma 1.** *Any walk between $x$ and $y$ with length $l \leq 2h + 1$ is included in $G_{x,y}^h$.*

*Proof.* Given any walk $w = \langle x, v_1, \cdots, v_{l-1}, y \rangle$ with length $l$, we will show that every node $v_i$ is included in $G_{x,y}^h$. Consider any $v_i$. Assume $d(v_i, x) \geq h + 1$ and $d(v_i, y) \geq h + 1$. Then, $2h + 1 \geq l = |\langle x, v_1, \cdots, v_i \rangle| + |\langle v_i, \cdots, v_{l-1}, y \rangle| \geq d(v_i, x) + d(v_i, y) \geq 2h + 2$, a contradiction. Thus, $d(v_i, x) \leq h$ or $d(v_i, y) \leq h$. By the definition of $G_{x,y}^h$, $v_i$ must be included in $G_{x,y}^h$. $\quad \square$

Next we will analyze Katz, rooted PageRank and SimRank one by one.

### 3.1 Katz index

The Katz index [29] for $(x, y)$ is defined as

$$\text{Katz}_{x,y} = \sum_{l=1}^{\infty} \beta^l |\text{walks}^{\langle l \rangle}(x, y)| = \sum_{l=1}^{\infty} \beta^l [A^l]_{x,y}, \tag{3}$$

where $\text{walks}^{\langle l \rangle}(x, y)$ is the set of length-$l$ walks between $x$ and $y$, and $A^l$ is the $l^{\text{th}}$ power of the adjacency matrix of the network. Katz index sums over the collection of all walks between $x$ and $y$ where a walk of length $l$ is damped by $\beta^l$ ($0 < \beta < 1$), giving more weight to shorter walks.

Katz index is directly defined in the form of a $\gamma$-decaying heuristic with $\eta = 1, \gamma = \beta$, and $f(x, y, l) = |\text{walks}^{\langle l \rangle}(x, y)|$. According to Lemma 1, $|\text{walks}^{\langle l \rangle}(x, y)|$ is calculable from $G_{x,y}^h$ for $l \leq 2h + 1$, thus property 2 in Theorem 2 is satisfied. Now we show when property 1 is satisfied.

**Proposition 1.** *For any nodes $i, j$, $[A^l]_{i,j}$ is bounded by $d^l$, where $d$ is the maximum node degree of the network.*

*Proof.* We prove it by induction. When $l = 1$, $A_{i,j} \leq d$ for any $(i, j)$. Thus the base case is correct. Now, assume by induction that $[A^l]_{i,j} \leq d^l$ for any $(i, j)$, we have

$$[A^{l+1}]_{i,j} = \sum_{k=1}^{|V|} [A^l]_{i,k} A_{k,j} \leq d^l \sum_{k=1}^{|V|} A_{k,j} \leq d^l d = d^{l+1}. \quad \square$$

Taking $\lambda = d$, we can see that whenever $d < 1/\beta$, the Katz index will satisfy property 1 in Theorem 2. In practice, the damping factor $\beta$ is often set to very small values like 5E-4 [1], which implies that Katz can be very well approximated from the $h$-hop enclosing subgraph.

## 3.2 PageRank

The rooted PageRank for node $x$ calculates the stationary distribution of a random walker starting at $x$, who iteratively moves to a random neighbor of its current position with probability $\alpha$ or returns to $x$ with probability $1 - \alpha$. Let $\pi_x$ denote the stationary distribution vector. Let $[\pi_x]_i$ denote the probability that the random walker is at node $i$ under the stationary distribution.

Let $P$ be the transition matrix with $P_{i,j} = \frac{1}{|\Gamma(v_j)|}$ if $(i,j) \in E$ and $P_{i,j} = 0$ otherwise. Let $\mathbf{e}_x$ be a vector with the $x^{\text{th}}$ element being 1 and others being 0. The stationary distribution satisfies

$$\pi_x = \alpha P \pi_x + (1 - \alpha)\mathbf{e}_x. \tag{4}$$

When used for link prediction, the score for $(x,y)$ is given by $[\pi_x]_y$ (or $[\pi_x]_y + [\pi_y]_x$ for symmetry). To show that rooted PageRank is a $\gamma$-decaying heuristic, we introduce the *inverse P-distance* theory [30], which states that $[\pi_x]_y$ can be equivalently written as follows:

$$[\pi_x]_y = (1 - \alpha) \sum_{w:x \rightsquigarrow y} P[w]\alpha^{\text{len}(w)}, \tag{5}$$

where the summation is taken over all walks $w$ starting at $x$ and ending at $y$ (possibly touching $x$ and $y$ multiple times). For a walk $w = \langle v_0, v_1, \cdots, v_k \rangle$, $\text{len}(w) := |\langle v_0, v_1, \cdots, v_k \rangle|$ is the length of the walk. The term $P[w]$ is defined as $\prod_{i=0}^{k-1} \frac{1}{|\Gamma(v_i)|}$, which can be interpreted as the probability of traveling $w$. Now we have the following theorem.

**Theorem 3.** *The rooted PageRank heuristic is a $\gamma$-decaying heuristic which satisfies the properties in Theorem 2.*

*Proof.* We first write $[\pi_x]_y$ in the following form.

$$[\pi_x]_y = (1 - \alpha) \sum_{l=1}^{\infty} \sum_{\substack{w:x \rightsquigarrow y \\ \text{len}(w)=l}} P[w]\alpha^l. \tag{6}$$

Defining $f(x,y,l) := \sum_{\substack{w:x \rightsquigarrow y \\ \text{len}(w)=l}} P[w]$ leads to the form of a $\gamma$-decaying heuristic. Note that $f(x,y,l)$ is the probability that a random walker starting at $x$ stops at $y$ with exactly $l$ steps, which satisfies $\sum_{z \in V} f(x,z,l) = 1$. Thus, $f(x,y,l) \leq 1 < \frac{1}{\alpha}$ (property 1). According to Lemma 1, $f(x,y,l)$ is also calculable from $G_{x,y}^h$ for $l \leq 2h+1$ (property 2). $\qquad\square$

## 3.3 SimRank

The SimRank score [10] is motivated by the intuition that two nodes are similar if their neighbors are also similar. It is defined in the following recursive way: if $x = y$, then $s(x,y) := 1$; otherwise,

$$s(x,y) := \gamma \frac{\sum_{a \in \Gamma(x)} \sum_{b \in \Gamma(y)} s(a,b)}{|\Gamma(x)| \cdot |\Gamma(y)|} \tag{7}$$

where $\gamma$ is a constant between 0 and 1. According to [10], SimRank has an equivalent definition:

$$s(x,y) = \sum_{w:(x,y) \multimap (z,z)} P[w]\gamma^{\text{len}(w)}, \tag{8}$$

where $w : (x,y) \multimap (z,z)$ denotes all simultaneous walks such that one walk starts at $x$, the other walk starts at $y$, and they first meet at **any** vertex $z$. For a simultaneous walk $w = \langle (v_0, u_0), \cdots, (v_k, u_k) \rangle$, $\text{len}(w) = k$ is the length of the walk. The term $P[w]$ is similarly defined as $\prod_{i=0}^{k-1} \frac{1}{|\Gamma(v_i)||\Gamma(u_i)|}$, describing the probability of this walk. Now we have the following theorem.

**Theorem 4.** *SimRank is a $\gamma$-decaying heuristic which satisfies the properties in Theorem 2.*

*Proof.* We write $s(x,y)$ as follows.

$$s(x,y) = \sum_{l=1}^{\infty} \sum_{\substack{w:(x,y) \multimap (z,z) \\ \text{len}(w)=l}} P[w]\gamma^l, \tag{9}$$

Defining $f(x,y,l) := \sum_{\substack{w:(x,y) \multimap (z,z) \\ \text{len}(w)=l}} P[w]$ reveals that SimRank is a $\gamma$-decaying heuristic. Note that $f(x,y,l) \leq 1 < \frac{1}{\gamma}$. It is easy to see that $f(x,y,l)$ is also calculable from $G_{x,y}^h$ for $l \leq h$. $\qquad\square$

**Discussion** There exist several other high-order heuristics based on path counting or random walk [6] which can be as well incorporated into the $\gamma$-decaying heuristic framework. We omit the analysis here. Our results reveal that most high-order heuristics inherently share the same $\gamma$-decaying heuristic form, and thus can be effectively approximated from an $h$-hop enclosing subgraph with exponentially smaller approximation error. We believe the ubiquity of $\gamma$-decaying heuristics is not by accident – it implies that a successful link prediction heuristic is better to put exponentially smaller weight on structures far away from the target, as remote parts of the network intuitively make little contribution to link existence. Our results build the foundation for learning heuristics from local subgraphs, as they imply that local enclosing subgraphs already **contain enough information to learn good graph structure features** for link prediction which is much desired considering learning from the entire network is often infeasible. To summarize, from the small enclosing subgraphs extracted around links, we are able to accurately calculate first and second-order heuristics, and approximate a wide range of high-order heuristics with small errors. Therefore, given adequate feature learning ability of the model used, learning from such enclosing subgraphs is expected to achieve performance at least as good as a wide range of heuristics. There is some related work which empirically verifies that local methods can often estimate PageRank and SimRank well [31, 32]. Another related theoretical work [33] establishes a condition of $h$ to achieve some fixed approximation error for ordinary PageRank.

## 4 SEAL: An implemetation of the theory using GNN

In this section, we describe our SEAL framework for link prediction. SEAL does not restrict the learned features to be in some particular forms such as $\gamma$-decaying heuristics, but instead learns general graph structure features for link prediction. It contains three steps: 1) enclosing subgraph extraction, 2) node information matrix construction, and 3) GNN learning. Given a network, we aim to learn automatically a "heuristic" that best explains the link formations. Motivated by the theoretical results, this function takes local enclosing subgraphs around links as input, and output how likely the links exist. To learn such a function, we train a graph neural network (GNN) over the enclosing subgraphs. Thus, the first step in SEAL is to extract enclosing subgraphs for a set of sampled positive links (observed) and a set of sampled negative links (unobserved) to construct the training data.

A GNN typically takes $(A, X)$ as input, where $A$ (with slight abuse of notation) is the adjacency matrix of the input enclosing subgraph, $X$ is the *node information matrix* each row of which corresponds to a node's feature vector. The second step in SEAL is to construct the node information matrix $X$ for each enclosing subgraph. This step is crucial for training a successful GNN link prediction model. In the following, we discuss this key step. The node information matrix $X$ in SEAL has three components: structural node labels, node embeddings and node attributes.

### 4.1 Node labeling

The first component in $X$ is each node's structural label. A node labeling is function $f_l : V \to \mathbb{N}$ which assigns an integer label $f_l(i)$ to every node $i$ in the enclosing subgraph. The purpose is to use different labels to **mark nodes' different roles** in an enclosing subgraph: 1) The center nodes $x$ and $y$ are the target nodes between which the link is located. 2) Nodes with different relative positions to the center have different structural importance to the link. A proper node labeling should mark such differences. If we do not mark such differences, GNNs will not be able to **tell where are the target nodes between which a link existence should be predicted**, and lose structural information.

Our node labeling method is derived from the following criteria: 1) The two target nodes $x$ and $y$ always have the distinctive label "1". 2) Nodes $i$ and $j$ have the same label if $d(i, x) = d(j, x)$ and $d(i, y) = d(j, y)$. The second criterion is because, intuitively, a node $i$'s topological position within an enclosing subgraph can be described by its *radius* with respect to the two center nodes, namely $(d(i, x), d(i, y))$. Thus, we let nodes on the same orbit have the same label, so that the node labels can reflect nodes' relative positions and structural importance within subgraphs.

Based on the above criteria, we propose a *Double-Radius Node Labeling* (DRNL) as follows. First, assign label 1 to $x$ and $y$. Then, for any node $i$ with $(d(i, x), d(i, y)) = (1, 1)$, assign label $f_l(i) = 2$. Nodes with radius $(1, 2)$ or $(2, 1)$ get label 3. Nodes with radius $(1, 3)$ or $(3, 1)$ get 4. Nodes with $(2, 2)$ get 5. Nodes with $(1, 4)$ or $(4, 1)$ get 6. Nodes with $(2, 3)$ or $(3, 2)$ get 7. So on and so forth. In other words, we iteratively assign larger labels to nodes with a larger radius w.r.t. both center nodes, where the label $f_l(i)$ and the double-radius $(d(i, x), d(i, y))$ satisfy

1) if $d(i,x) + d(i,y) \neq d(j,x) + d(j,y)$, then $d(i,x) + d(i,y) < d(j,x) + d(j,y) \Leftrightarrow f_l(i) < f_l(j)$;

2) if $d(i,x) + d(i,y) = d(j,x) + d(j,y)$, then $d(i,x)d(i,y) < d(j,x)d(j,y) \Leftrightarrow f_l(i) < f_l(j)$.

One advantage of DRNL is that it has a perfect hashing function

$$f_l(i) = 1 + min(d_x, d_y) + (d/2)[(d/2) + (d\%2) - 1], \qquad (10)$$

where $d_x := d(i,x)$, $d_y := d(i,y)$, $d := d_x + d_y$, $(d/2)$ and $(d\%2)$ are the integer quotient and remainder of $d$ divided by 2, respectively. This perfect hashing allows fast closed-form computations.

For nodes with $d(i,x) = \infty$ or $d(i,y) = \infty$, we give them a null label 0. Note that DRNL is not the only possible way of node labeling, but we empirically verified its better performance than no labeling and other naive labelings. We discuss more about node labeling in Appendix B. After getting the labels, we use their one-hot encoding vectors to construct $X$.

## 4.2 Incorporating latent and explicit features

Other than the structural node labels, the node information matrix $X$ also provides an opportunity to include latent and explicit features. By concatenating each node's embedding/attribute vector to its corresponding row in $X$, we can make SEAL simultaneously learn from all three types of features.

Generating the node embeddings for SEAL is nontrivial. Suppose we are given the observed network $G = (V, E)$, a set of sampled positive training links $E_p \subseteq E$, and a set of sampled negative training links $E_n$ with $E_n \cap E = \varnothing$. If we directly generate node embeddings on $G$, the node embeddings will record the link existence information of the training links (since $E_p \subseteq E$). We observed that GNNs can quickly find out such link existence information and optimize by only fitting this part of information. This results in bad generalization performance in our experiments. Our trick is to temporally add $E_n$ into $E$, and generate the embeddings on $G' = (V, E \cup E_n)$. This way, the positive and negative training links will have the same link existence information recorded in the embeddings, so that GNN cannot classify links by only fitting this part of information. We empirically verified the much improved performance of this trick to SEAL. We name this trick *negative injection*.

We name our proposed framework **SEAL** (learning from Subgraphs, Embeddings and Attributes for Link prediction), emphasizing its ability to jointly learn from three types of features.

## 5 Experimental results

We conduct extensive experiments to evaluate SEAL. Our results show that SEAL is a superb and robust framework for link prediction, achieving unprecedentedly strong performance on various networks. We use AUC and average precision (AP) as evaluation metrics. We run all experiments for 10 times and report the average AUC results and standard deviations. We leave the the AP and time results in Appendix F. SEAL is flexible with what GNN or node embeddings to use. Thus, we choose a recent architecture DGCNN [17] as the default GNN, and node2vec [20] as the default embeddings. The code and data are available at https://github.com/muhanzhang/SEAL.

**Datasets**   The eight datasets used are: USAir, NS, PB, Yeast, C.ele, Power, Router, and E.coli (please see Appendix C for details). We randomly remove 10% existing links from each dataset as positive testing data. Following a standard manner of learning-based link prediction, we randomly sample the same number of nonexistent links (unconnected node pairs) as negative testing data. We use the remaining 90% existing links as well as the same number of additionally sampled nonexistent links to construct the training data.

**Comparison to heuristic methods**   We first compare SEAL with methods that only use graph structure features. We include eight popular heuristics (shown in Appendix A, Table 3): common neighbors (CN), Jaccard, preferential attachment (PA), Adamic-Adar (AA), resource allocation (RA), Katz, PageRank (PR), and SimRank (SR). We additionally include Ensemble (ENS) which trains a logistic regression classifier on the eight heuristic scores. We also include two heuristic learning methods: Weisfeiler-Lehman graph kernel (WLK) [34] and WLNM [12], which also learn from (truncated) enclosing subgraphs. We omit path ranking methods [28] as well as other recent methods which are specifically designed for knowledge graphs or recommender systems [23, 35]. As all the baselines only use graph structure features, we restrict SEAL to not include any latent or explicit features. In SEAL, the hop number $h$ is an important hyperparameter. Here, we select $h$ only from $\{1, 2\}$, since on one hand we empirically verified that the performance typically does not increase

after $h \geq 3$, which validates our theoretical results that the most useful information is within local structures. On the other hand, even $h = 3$ sometimes results in very large subgraphs if a hub node is included. This raises the idea of sampling nodes in subgraphs, which we leave to future work. The selection principle is very simple: If the second-order heuristic AA outperforms the first-order heuristic CN on 10% validation data, then we choose $h = 2$; otherwise we choose $h = 1$. For datasets PB and E.coli, we consistently use $h = 1$ to fit into the memory. We include more details about the baselines and hyperparameters in Appendix D.

**Table 1:** Comparison with heuristic methods (AUC).

| Data | CN | Jaccard | PA | AA | RA | Katz | PR | SR | ENS | WLK | WLNM | SEAL |
|---|---|---|---|---|---|---|---|---|---|---|---|---|
| USAir | 93.80±1.22 | 89.79±1.61 | 88.84±1.45 | 95.06±1.03 | 95.77±0.92 | 92.88±1.42 | 94.67±1.08 | 78.89±2.31 | 88.96±1.44 | **96.63**±0.73 | 95.95±1.10 | **96.62**±0.72 |
| NS | 94.42±0.95 | 94.43±0.93 | 68.65±2.03 | 94.45±0.93 | 94.45±0.93 | 94.85±1.10 | 94.89±1.08 | 94.79±1.08 | 97.64±0.25 | 98.57±0.51 | 98.61±0.49 | **98.85**±0.47 |
| PB | 92.04±0.35 | 87.41±0.39 | 90.14±0.45 | 92.36±0.34 | 92.46±0.37 | 92.92±0.35 | 93.54±0.41 | 77.08±0.80 | 90.15±0.45 | 93.83±0.59 | 93.49±0.47 | **94.72**±0.46 |
| Yeast | 89.37±0.61 | 89.32±0.60 | 82.20±1.02 | 89.43±0.62 | 89.45±0.62 | 92.24±0.61 | 92.76±0.55 | 91.49±0.57 | 82.36±1.02 | 95.86±0.54 | 95.62±0.52 | **97.91**±0.52 |
| C.ele | 85.13±1.61 | 80.19±1.64 | 74.79±2.04 | 86.95±1.40 | 87.49±1.41 | 86.34±1.89 | **90.32**±1.49 | 77.07±2.00 | 74.94±2.04 | 89.72±1.67 | 86.18±1.72 | **90.30**±1.35 |
| Power | 58.80±0.88 | 58.79±0.88 | 44.33±1.02 | 58.79±0.88 | 58.79±0.88 | 65.39±1.59 | 66.00±1.59 | 76.15±1.06 | 79.52±1.78 | 82.41±3.43 | 84.76±0.98 | **87.61**±1.57 |
| Router | 56.43±0.52 | 56.40±0.52 | 47.58±1.47 | 56.43±0.51 | 56.43±0.51 | 38.62±1.35 | 38.76±1.39 | 37.40±1.27 | 47.58±1.48 | 87.42±2.08 | 94.41±0.88 | **96.38**±1.45 |
| E.coli | 93.71±0.39 | 81.31±0.61 | 91.82±0.58 | 95.36±0.34 | 95.95±0.35 | 93.50±0.44 | 95.57±0.44 | 62.49±1.43 | 91.89±0.58 | 96.94±0.29 | 97.21±0.27 | **97.64**±0.22 |

Table 1 shows the results. Firstly, we observe that methods which learn from enclosing subgraphs (WLK, WLNM and SEAL) generally perform much better than predefined heuristics. This indicates that the learned "heuristics" are better at capturing the network properties than manually designed ones. Among learning-based methods, SEAL has the best performance, demonstrating GNN's superior graph feature learning ability over graph kernels and fully-connected neural networks. From the results on Power and Router, we can see that although existing heuristics perform similarly to random guess, learning-based methods still maintain high performance. This suggests that we can even discover new "heuristics" for networks where no existing heuristics work.

**Table 2:** Comparison with latent feature methods (AUC).

| Data | MF | SBM | N2V | LINE | SPC | VGAE | SEAL |
|---|---|---|---|---|---|---|---|
| USAir | 94.08±0.80 | 94.85±1.14 | 91.44±1.78 | 81.47±10.71 | 74.22±3.11 | 89.28±1.99 | **97.09**±0.70 |
| NS | 74.55±4.34 | 92.30±2.26 | 91.52±1.28 | 80.63±1.90 | 89.94±2.39 | 94.04±1.64 | **97.71**±0.93 |
| PB | 94.30±0.53 | 93.90±0.42 | 85.79±0.78 | 76.95±2.76 | 83.96±0.86 | 90.70±0.53 | **95.01**±0.34 |
| Yeast | 90.28±0.69 | 91.41±0.60 | 93.67±0.46 | 87.45±3.33 | 93.25±0.40 | 93.88±0.21 | **97.20**±0.64 |
| C.ele | 85.90±1.74 | 86.48±2.60 | 84.11±1.27 | 69.21±3.14 | 51.90±2.57 | 81.80±2.18 | **89.54**±2.04 |
| Power | 50.63±1.10 | 66.57±2.05 | 76.22±0.92 | 55.63±1.47 | **91.78**±0.61 | 71.20±1.65 | 84.18±1.82 |
| Router | 78.03±1.63 | 85.65±1.93 | 65.46±0.86 | 67.15±2.10 | 68.79±2.42 | 61.51±1.22 | **95.68**±1.22 |
| E.coli | 93.76±0.56 | 93.82±0.41 | 90.82±1.49 | 82.38±2.19 | 94.92±0.32 | 90.81±0.63 | **97.22**±0.28 |

**Comparison to latent feature methods** Next we compare SEAL with six state-of-the-art latent feature methods: matrix factorization (MF), stochastic block model (SBM) [18], node2vec (N2V) [20], LINE [21], spectral clustering (SPC), and variational graph auto-encoder (VGAE) [36]. Among them, VGAE uses a GNN too. Please note the difference between VGAE and SEAL: VGAE uses a node-level GNN to learn node embeddings that best reconstruct the network, while SEAL uses a graph-level GNN to classify enclosing subgraphs. Therefore, VGAE still belongs to latent feature methods. For SEAL, we additionally include the 128-dimensional node2vec embeddings in the node information matrix $X$. Since the datasets do not have node attributes, explicit features are not included.

Table 2 shows the results. As we can see, SEAL shows significant improvement over latent feature methods. One reason is that SEAL learns from both graph structures and latent features simultaneously, thus augmenting those methods that only use latent features. We observe that SEAL with node2vec embeddings outperforms pure node2vec by large margins. This implies that network embeddings alone may not be able to capture the most useful link prediction information located in the local structures. It is also interesting that compared to SEAL without node2vec embeddings (Table 1), joint learning does not always improve the performance. More experiments and discussion are included in Appendix F.

# 6 Conclusions

Learning link prediction heuristics automatically is a new field. In this paper, we presented theoretical justifications for learning from local enclosing subgraphs. In particular, we proposed a $\gamma$-decaying theory to unify a wide range of high-order heuristics and prove their approximability from local subgraphs. Motivated by the theory, we proposed a novel link prediction framework, SEAL, to simultaneously learn from local enclosing subgraphs, embeddings and attributes based on graph neural networks. Experimentally we showed that SEAL achieved unprecedentedly strong performance by comparing to various heuristics, latent feature methods, and network embedding algorithms. We hope SEAL can not only inspire link prediction research, but also open up new directions for other relational machine learning problems such as knowledge graph completion and recommender systems.

**Acknowledgments**

The work is supported in part by the III-1526012 and SCH-1622678 grants from the National Science Foundation and grant 1R21HS024581 from the National Institute of Health.

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
