[Supplementary Material]

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

# Appendices

## A  More about the three types of features for link prediction

In this section, we discuss more about the difference among the three types commonly used features for link prediction: graph structure features, latent features, and explicit features.

**Graph structure features** locate inside the observed node and edge structures of the network, which can be directly observed and computed. Link prediction heuristics belong to graph structure features. We show eight popular heuristics in Table 3. In addition to link prediction heuristics, node centrality scores (degree, closeness, betweenness, PageRank, eigenvector, hubs etc.), graphlets, network motifs etc. all belong to graph structure features. Although effective in many domains, these predefined graph structure features are handcrafted – they only capture a small set of structure patterns, lacking the ability to express general structure patterns underlying different networks. Considering deep neural networks' success in feature learning, a natural question to ask is whether we can automatically learn such features, no longer relying on predefined ones.

Graph structure features are inductive, meaning that these features are not associated with a particular node or network. For example, the common neighbor heuristic between any pair of nodes $x$ and $y$ is consistently calculated by counting the number of their common one-hop neighbors, invariant to where $x$ and $y$ are located. Thus, graph structure features are transferrable to new nodes and new networks. This is in contrast to latent features, which are often transductive – the changing of network structure will require a complete retraining to get the latent features again.

**Table 3:** Popular heuristics for link prediction, see [1] for details.

| Name | Formula | Order |
|---|---|---|
| common neighbors | $\lvert \Gamma(x) \cap \Gamma(y) \rvert$ | first |
| Jaccard | $\frac{\lvert \Gamma(x) \cap \Gamma(y) \rvert}{\lvert \Gamma(x) \cup \Gamma(y) \rvert}$ | first |
| preferential attachment | $\lvert \Gamma(x) \rvert \cdot \lvert \Gamma(y) \rvert$ | first |
| Adamic-Adar | $\sum_{z \in \Gamma(x) \cap \Gamma(y)} \frac{1}{\log \lvert \Gamma(z) \rvert}$ | second |
| resource allocation | $\sum_{z \in \Gamma(x) \cap \Gamma(y)} \frac{1}{\lvert \Gamma(z) \rvert}$ | second |
| Katz | $\sum_{l=1}^{\infty} \beta^l \lvert \text{walks}^{\langle l \rangle}(x,y) \rvert$ | high |
| PageRank | $[\pi_x]_y + [\pi_y]_x$ | high |
| SimRank | $\gamma \frac{\sum_{a \in \Gamma(x)} \sum_{b \in \Gamma(y)} \text{score}(a,b)}{\lvert \Gamma(x) \rvert \cdot \lvert \Gamma(y) \rvert}$ | high |

Notes: $\Gamma(x)$ denotes the neighbor set of vertex $x$. $\beta < 1$ is a damping factor. $\lvert \text{walks}^{\langle l \rangle}(x,y) \rvert$ counts the number of length-$l$ walks between $x$ and $y$. $[\pi_x]_y$ is the stationary distribution probability of $y$ under the random walk from $x$ with restart, see [9]. SimRank score is a recursive definition. We exclude those heuristics which are simple variants of the above or are proven to be meaningless for large graphs (e.g., commute time [37]).

**Latent features** are latent properties or representations of nodes, often obtained by factorizing a specific matrix derived from a network, such as the adjacency matrix or the Laplacian matrix. Through factorization, a low-dimensional embedding is learned for each node. Latent features focus more on global properties and long range effects, because the network's matrix is treated as a whole during factorization. Latent features cannot capture structural similarities between nodes [38], and usually need an extremely large dimension to express some simple heuristics [23]. Latent features are also transductive. They cannot be transferred to new nodes or new networks. They are also less interpretable than graph structure features.

Network embedding methods [19, 21, 20, 39, 40, 41] have gained great popularity recently. They learn low-dimensional representations for nodes too. Recently, it is shown that network embedding methods (including DeepWalk [19], LINE [21], and node2vec [20]) implicitly factorize some matrix representation of a network [22]. For example, DeepWalk approximately factorizes $\log(\text{vol}(G)(\frac{1}{T}\sum_{r=1}^{T}(D^{-1}A)^r)D^{-1}) - \log(b)$, where $A$ is the adjacency matrix of the network $G$, $D$ is the diagonal degree matrix, $T$ is skip-gram's window size, and $b$ is the number of negative

**Figure 2:** Double-Radius Node Labeling.

samples. For LINE and node2vec, there also exist such matrices. Since network embedding methods also factorize matrix representations of networks, we may regard them as learning more expressive latent features through factorizing some more informative matrices.

**Explicit features** are often given by continuous or discrete node attribute vectors. In principle, any side information about the network other than its structure can be seen as explicit features. For example, in citation networks, word distributions are explicit features of document nodes. In social networks, a user's profile information is also explicit feature (however, their friendship information belongs to graph structure features).

These three types of features are largely orthogonal to each other. Many papers have considered using them together for link prediction [42, 43, 23, 24] to improve the performance of single-feature-based methods.

## B More discussion about node labeling

The necessity of structural node labels for enclosing subgraphs is because, unlike ordinary graphs, enclosing subgraphs intrinsically have a directionality. The center of an enclosing subgraph are two nodes $x$ and $y$ between which the target link is located. Outward from the center, other nodes have larger and larger distance to $x$ and $y$. Node labeling is to mark such structural differences thus providing additional structural information to facilitate GNN training.

When designing a node labeling for enclosing subgraphs, we always want to ensure that the target nodes $x$ and $y$ have a distinct label so that GNN can distinguish the target link to predict from other edges. Secondly, we want the node labels to reflect nodes' relative positions in their enclosing subgraph. This relative position can be intuitively described by a node $i$'s double-radius with respect to $x$ and $y$, i.e., $(d(i,x), d(i,y))$.

We restate our Double-Radius Node Labeling (DRNL) algorithm here. First, assign label 1 to $x$ and $y$. Then, for any node $i$ with $(d(i,x), d(i,y)) = (1,1)$, assign label $f_l(i) = 2$. Nodes with double-radius $(1,2)$ or $(2,1)$ get label 3. Nodes with double-radius $(1,3)$ or $(3,1)$ get label 4. Nodes with $(2,2)$ get label 5. Nodes with $(1,4)$ or $(4,1)$ get label 6. Nodes with $(2,3)$ or $(3,2)$ get label 7. So on and so forth. Our DRNL not only satisfies the above criteria, but also attains the additional benefits that for nodes $i$ and $j$:

1) if $d(i,x) + d(i,y) \neq d(j,x) + d(j,y)$, then $d(i,x) + d(i,y) < d(j,x) + d(j,y) \Leftrightarrow f_l(i) < f_l(j)$;

2) if $d(i,x) + d(i,y) = d(j,x) + d(j,y)$, then $d(i,x)d(i,y) < d(j,x)d(j,y) \Leftrightarrow f_l(i) < f_l(j)$.

That is, the magnitude of node labels also reflects their distance to the center. Nodes with smaller arithmetic mean distance to the target nodes get smaller labels. If two nodes have the same arithmetic mean distance, the node with a smaller geometric mean distance to the target nodes gets a smaller label. Note that these additional benefits will not be available under one-hot encoding of node labels, since the magnitude information will be lost after one-hot encoding. However, such a labeling is potentially useful when node labels are directly used for training, or used to rank the nodes. Furthermore, our node labeling has a perfect hashing (10) which allows closed-form computation.

We present a lookup table for DRNL and an example labeled subgraph in Figure 2. Note that when calculating $d(i,x)$, we temporally remove $y$ from the subgraph, and vice versa. This is because we aim to use the pure distance between $i$ and $x$ without the influence of $y$. If we do not remove $y$, $d(i,x)$ will be upper bounded by $d(i,y) + d(x,y)$, obscuring the "true distance" between $i$ and $x$.

Our node labeling algorithm is different from the Weisfeiler-Lehman algorithm used in WLNM [12]. In WLNM, node labeling is for defining a node order in adjacency matrices – the labels are not really

input to machine learning models. To rank nodes with least ties, the node labels should be as fine as possible in WLNM. In comparison, the node labels in SEAL need not be very fine, as their purpose is for indicating nodes' different roles within the enclosing subgraph, not for ranking nodes. In addition, node labels in SEAL are encoded into node information matrices and input to machine learning models.

## C  Dataset details

USAir [44] is a network of US Air lines with 332 nodes and 2,126 edges. The average node degree is 12.81. NS [45] is a collaboration network of researchers in network science with 1,589 nodes and 2,742 edges. The average node degree is 3.45. PB [46] is a network of US political blogs with 1,222 nodes and 16,714 edges. The average node degree is 27.36. Yeast [47] is a protein-protein interaction network in yeast with 2,375 nodes and 11,693 edges. The average node degree is 9.85. C.ele [48] is a neural network of C. elegans with 297 nodes and 2,148 edges. The average node degree is 14.46. Power [48] is an electrical grid of western US with 4,941 nodes and 6,594 edges. The average node degree is 2.67. Router [49] is a router-level Internet with 5,022 nodes and 6,258 edges. The average node degree is 2.49. E.coli [50] is a pairwise reaction network of metabolites in E. coli with 1,805 nodes and 14,660 edges. The average node degree is 12.55.

## D  Additional details about baselines

**Hyperparameters of heuristic and latent feature methods**  Most hyperparameters are inherited from the original paper of each method. For Katz, we set the damping factor $\beta$ to 0.001. For PageRank, we set the damping factor $\alpha$ to 0.85. For SimRank, we set $\gamma$ to 0.8. For stochastic block model (SBM), we use the implementation of [51] using a latent group number 12. For matrix factorization (MF), we use the libFM [52] software with the default parameters. For node2vec, LINE, and spectral clustering, we first generate 128-dimensional embeddings from the observed networks with default parameters of each software. Then, we use the Hadamard product of two nodes' embeddings as a link's embedding as suggested in [20], and train a logistic regression model with Liblinear [53] using automatic hyperparameter selection. For VGAE, we use its default setting.

**WLNM**  Weisfeiler-Lehman Neural Machine (WLNM) [12] is a recent link prediction method that learns general graph structure features. It achieves state-of-the-art performance on various networks, outperforming all handcrafted heuristics. WLNM has three steps: enclosing subgraph extraction, subgraph pattern encoding, and neural network training. In the enclosing subgraph extraction step: for each node pair $(x, y)$, WLNM iteratively extracts $x$ and $y$'s one-hop neighbors, two-hop neighbors, and so on, until the enclosing subgraph has **more than** $K$ vertices, where $K$ is a user-defined integer. In the subgraph pattern encoding step, WLNM uses the Weisfeiler-Lehman algorithm to define an order for nodes within each enclosing subgraph, so that the neural network can read different subgraphs' nodes in a consistent order and learn meaningful patterns. To unify the sizes of the enclosing subgraphs, after getting the vertex order, the last few vertices are deleted so that all the truncated enclosing subgraphs have the same size $K$. These truncated enclosing subgraphs are reordered and their fixed-size adjacency matrices are fed into the fully-connected neural network to train a link prediction model. Due to the truncation, WLNM cannot consistently learn from each link's full $h$-hop neighborhood. The loss of structural information limits WLNM's performance and restrict it from learning complete $h$-order graph structure features. Following [12], we use $K = 10$ (the best performing $K$) in our experiments.

**Table 4:** Comparison of different link prediction methods

|  | Heuristics | Latent features | WLK | WLNM | SEAL |
|---|---|---|---|---|---|
| Graph structure features | Yes | No | Yes | Yes | Yes |
| Learn from full $h$-hop | No | n/a | Yes | No | Yes |
| Latent/explicit features | No | Yes | No | No | Yes |
| Model | n/a | LR/inner product | SVM | NN | GNN |

**WLK**  Weisfeiler-Lehman graph kernel (WLK) [34] is a state-of-the-art graph kernel. Graph kernels make kernel machines feasible for graph classification by defining some positive semidefinite graph similarity scores. Most graph kernels measure graph similarity by decomposing graphs into small

**Figure 3:** The DGCNN architecture.

substructures and adding up the pair-wise similarities between these components. Common types of substructures include walks [54, 55], subgraphs [56, 57], paths [58], and subtrees [34, 59]. WLK is based on counting common rooted subtrees between two graphs. In our experiments, we train a SVM on the WL kernel matrix. We feed the same enclosing subgraphs as in SEAL to WLK. We search the subtree depth from $\{0, 1, 2, 3, 4, 5\}$ on 10% validation links. WLK does not support continuous node information, but supports integer node labels. Thus, we feed the same structural node labels from (10) to WLK too.

We compare the characteristics of different link prediction methods in Table 4.

## E  Configuration details of SEAL

In the experiments, we use Deep Graph Convolutional Neural Network (DGCNN) [17] as the default GNN engine of SEAL. DGCNN is a recent GNN architecture for graph classification. It has consistently good performance on various benchmark datasets with a single network architecture (avoid hyperparameter tweaking). DGCNN is equipped with propagation-based graph convolution layers and a novel graph aggregation layer, called SortPooling. We illustrate the overall architecture of DGCNN in Figure 3. Given the adjacency matrix $A \in \{1, 0\}^{n \times n}$ and the node information matrix $X \in \mathbb{R}^{n \times c}$ of an enclosing subgraph, DGCNN uses the following graph convolution layer:

$$Z = f(\tilde{D}^{-1}\tilde{A}XW), \tag{11}$$

where $\tilde{A} = A + I$, $\tilde{D}$ is a diagonal degree matrix with $\tilde{D}_{i,i} = \sum_j \tilde{A}_{i,j}$, $W \in \mathbb{R}^{c \times c'}$ is a matrix of trainable graph convolution parameters, $f$ is an element-wise nonlinear activation function, and $Z \in \mathbb{R}^{n \times c'}$ are the new node states. The mechanism behind (11) is that the initial node states $X$ are first applied a linear transformation by multiplying $W$, and then propagated to neighboring nodes through the propagation matrix $\tilde{D}^{-1}\tilde{A}$. After graph convolution, the $i^{\text{th}}$ row of $Z$ becomes:

$$Z_i = f\big(\frac{1}{|\Gamma(i)| + 1}[X_iW + \sum_{j \in \Gamma(i)} X_jW]\big), \tag{12}$$

which summarizes the node information as well as the first-order structure pattern from $i$'s neighbors. DGCNN stacks multiple graph convolution layers (11) and concatenates each layer's node states as the final node states, in order to extract multi-hop node features.

A graph aggregation layer constructs a graph-level feature vector from individual nodes' final states, which is used for graph classification. The most widely used aggregation operation is summing, i.e., nodes' final states after graph convolutions are summed up as the graph's representation. However, the averaging effect of summing might lose much individual nodes' information as well as the topological information of the graph. DGCNN uses a novel SortPooling layer, which sorts the final node states according to the last graph convolution layer's output to achieve an isomorphism invariant node ordering [17]. A max-$k$ pooling operation is then used to unify the sizes of the sorted representations of different graphs, which enables training a traditional 1-D CNN on the node sequence.

We use the default setting of DGCNN, i.e., four graph convolution layers as in (11) with 32,32,32,1 channels, a SortPooling layer (with $k$ such that 60% graphs have nodes less than $k$), two 1-D convolution layers (16 and 32 output channels) and a dense layer (128 neurons), see [17]. We train DGCNN on enclosing subgraphs for 50 epochs, and select the model with the smallest loss on the 10% validation data to predict the testing links.

Note that, in any positive training link's enclosing subgraph, we should always remove the edge between the two target nodes before feeding it into a graph classification model. This is because this edge will contain the link existence information, which is not available in any testing link's enclosing subgraph.

# F   Additional results

In this section, we show the additional experimental results. We first use 90% observed links as training links and 10% as testing links following the main paper's experiments. The average precision (AP) comparison results with heuristic methods are shown in Table 5. The AP comparison results with latent feature methods are shown in Table 6. We can see that our proposed SEAL shows great performance improvement over all baselines in both AUC and AP.

**Table 5:** Comparison with heuristic methods (AP), 90% training links.

| Data | CN | Jaccard | PA | AA | RA | Katz | PR | SR | ENS | WLK | WLNM | SEAL |
|---|---|---|---|---|---|---|---|---|---|---|---|---|
| USAir | 93.45±1.19 | 87.54±2.07 | 91.22±1.28 | 95.36±1.00 | 96.27±0.79 | 94.07±1.18 | 95.08±1.16 | 69.24±2.61 | 91.33±1.27 | **96.82**±0.84 | 95.95±1.13 | **96.80**±0.55 |
| NS | 94.39±0.96 | 94.44±0.93 | 72.85±1.88 | 94.46±0.93 | 94.46±0.93 | 95.05±1.08 | 95.11±1.04 | 94.98±1.02 | 97.68±0.36 | 98.79±0.40 | 98.81±0.49 | **99.06**±0.37 |
| PB | 91.47±0.45 | 84.78±0.71 | 89.33±0.72 | 92.36±0.46 | 92.37±0.57 | 93.07±0.46 | 92.97±0.77 | 64.33±0.95 | 89.35±0.71 | 93.34±0.89 | 92.69±0.64 | **94.31**±0.56 |
| Yeast | 89.34±0.62 | 89.15±0.67 | 85.36±0.85 | 89.53±0.63 | 89.55±0.63 | 95.23±0.39 | 95.47±0.43 | 93.42±0.64 | 85.54±0.85 | 96.82±0.35 | 96.40±0.38 | **98.33**±0.37 |
| C.ele | 82.62±1.51 | 77.06±2.55 | 75.49±1.86 | 86.46±1.43 | 87.10±1.53 | 85.93±1.69 | **89.56**±1.57 | 68.61±2.31 | 75.69±1.86 | 88.96±2.06 | 85.08±2.05 | 89.48±1.85 |
| Power | 58.77±0.88 | 58.77±0.89 | 51.93±1.16 | 58.76±0.89 | 58.76±0.90 | 79.82±0.91 | 80.56±0.91 | 77.02±0.93 | 83.63±1.37 | 83.02±3.19 | 87.16±0.77 | **89.55**±1.29 |
| Router | 56.39±0.53 | 55.84±0.80 | 69.03±0.95 | 56.50±0.51 | 56.51±0.50 | 64.52±0.81 | 64.91±0.85 | 58.82±1.12 | 69.25±0.96 | 86.59±2.23 | 93.53±1.09 | **96.23**±1.71 |
| E.coli | 93.49±0.38 | 82.42±0.59 | 94.04±0.33 | 96.05±0.25 | 96.72±0.25 | 94.83±0.30 | 96.41±0.33 | 55.01±0.86 | 94.11±0.33 | 97.25±0.42 | 97.50±0.23 | **98.03**±0.20 |

**Table 6:** Comparison with latent feature methods (AP), 90% training links.

| Data | MF | SBM | N2V | LINE | SPC | VGAE | SEAL |
|---|---|---|---|---|---|---|---|
| USAir | 94.36±0.79 | 95.08±1.10 | 89.71±2.97 | 79.70±11.76 | 78.07±2.92 | 89.27±1.29 | **97.13**±0.80 |
| NS | 78.41±3.85 | 92.13±2.36 | 94.28±0.91 | 85.17±1.65 | 90.83±2.16 | 95.83±1.04 | **98.12**±0.77 |
| PB | 93.56±0.71 | 93.35±0.52 | 84.79±1.03 | 78.82±2.71 | 86.57±0.61 | 90.38±0.72 | **94.55**±0.43 |
| Yeast | 92.01±0.47 | 92.73±0.44 | 94.90±0.38 | 90.55±2.39 | 94.63±0.56 | 95.19±0.36 | **97.95**±0.35 |
| C.ele | 83.63±2.09 | 84.66±2.95 | 83.12±1.90 | 67.51±2.72 | 62.07±2.40 | 78.32±3.49 | **88.81**±2.32 |
| Power | 53.50±1.22 | 65.48±1.85 | 81.49±0.86 | 56.66±1.43 | **91.00**±0.58 | 75.91±1.56 | 86.69±1.50 |
| Router | 82.59±1.38 | 84.67±1.89 | 68.66±1.49 | 71.92±1.53 | 73.53±1.47 | 70.36±0.85 | **95.66**±1.23 |
| E.coli | 95.59±0.31 | 95.30±0.27 | 90.87±1.48 | 86.45±1.82 | 96.08±0.37 | 92.77±0.65 | **97.83**±0.20 |

To evaluate SEAL's scalability, we show its single-GPU inference time performance in Table 7. As we can see, SEAL has good scalability. For networks with over 1E7 potential links, SEAL took less than an hour to make all the predictions. One possible way to further scale SEAL to social networks with millions of users is to first use some simple heuristics such as common neighbors to filter out most unlikely links and then use SEAL to make further recommendations. Another way is to restrict the candidate friend recommendations to be those who are at most 2 or 3 hops away from the target user, which will vastly reduce the number of candidate links to infer for each user and thus further increase the scalability.

**Table 7:** Inference time of SEAL.

| | USAir | NS | PB | Yeast | C.ele | Power | Router | E.coli |
|---|---|---|---|---|---|---|---|---|
| Number of potential links | 5.49E+04 | 1.26E+06 | 7.46E+05 | 2.82E+06 | 4.40E+04 | 1.22E+07 | 1.26E+07 | 1.39E+06 |
| Inference time per link (s) | 6.05E-04 | 2.55E-04 | 2.04E-04 | 3.96E-04 | 4.13E-04 | 1.35E-04 | 2.13E-04 | 2.40E-04 |
| Inference time for all potential links (s) | 31 | 321 | 146 | 1106 | 16 | 1640 | 2681 | 328 |

Next, we redo the comparisons under 50%–50% train/test split. We randomly remove 50% existing links as positive testing links and use the remaining 50% existing links as positive training links. The same number of negative training and testing links are sampled from the nonexistent links as well. The AUC results are shown in Table 8 and 9. The AP results are shown in Table 10 and 11.

The results are consistent with the 90%–10% split setting. As we can see, SEAL is still the best among all methods in general. The performance gains over heuristic methods are even larger compared to the 90%-10% split. This indicates that SEAL is able to learn good heuristics even when the network is very incomplete. SEAL also shows more clear advantages over WLNM. On the other hand, we observe that VGAE becomes a strong baseline when network is sparser by achieving the best AUC results on 3 out of 8 datasets. It is thus interesting to study whether replacing the node2vec embeddings in SEAL with the VGAE embeddings can further improve the performance. We leave it to future work.

**Table 8:** Comparison with heuristic methods (AUC), 50% training links.

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

We further conduct experiments with the setting of the node2vec paper [20] on five networks: arXiv (18,722 nodes and 198,110 edges) [60], Facebook (4,039 nodes and 88,234 edges) [60], BlogCatalog (10,312 nodes, 333,983 edges and 39 attributes) [61], Wikipedia (4,777 nodes, 184,812 edges and 40 attributes) [62], and Protein-Protein Interactions (PPI) (3,890 nodes, 76,584 edges and 50 attributes) [63]. For each network, 50% of random links are removed and used as testing data, while keeping the remaining network connected. For Facebook and arXiv, all remained links are used as positive training data. For PPI, BlogCatalog and Wikipedia, we sample 10,000 remained links as positive training data. We compare SEAL ($h = 1$, 10 training epochs) with node2vec, LINE, SPC, VGAE, and WLNM ($K = 10$). For node2vec, we use the parameters provided in [20] if available. For SEAL and VGAE, the node attributes are used since only these two methods support explicit features.

Table 12 shows the results. As we can see, SEAL consistently outperforms all embedding methods. Especially on the last three networks, SEAL (with node2vec embeddings) outperforms pure node2vec by large margins. These results indicate that in many cases, embedding methods alone cannot capture the most useful link prediction information, while effectively combining the power of different types of features results in much better performance. SEAL also consistently outperforms WLNM.

**Table 12:** Comparison with network embedding methods (AUC and standard deviation, OOM: out of memory).

| | N2V | LINE | SPC | VGAE | WLNM | SEAL |
|---|---|---|---|---|---|---|
| arXiv | 96.18±0.40 | 84.64±0.03 | 87.00±0.14 | OOM | 99.19±0.03 | **99.40**±0.14 |
| Facebook | 99.05±0.07 | 89.63±0.06 | 98.59±0.11 | 98.21±0.22 | 99.24±0.03 | **99.40**±0.08 |
| BlogCatalog | 85.97±1.56 | 90.92±2.05 | 96.74±0.31 | OOM | 96.55±0.08 | **98.10**±0.60 |
| Wikipedia | 76.59±2.06 | 74.44±0.66 | 99.54±0.04 | 89.74±0.18 | 99.05±0.03 | **99.63**±0.05 |
| PPI | 70.31±0.79 | 72.82±1.53 | 92.27±0.22 | 85.86±0.43 | 88.79±0.38 | **93.52**±0.37 |