[Reviews · NeurIPS 2018]

Reviewer 1



Quality: From the technical point of view, the paper seems to be well-prepared. In particular, the authors propose $\gamma-heuristic theory$ for link prediction and show that many common heuristics can be expressed in a general formulation. They build their framework based on this theory and on the previous work (WLNM). However, I would like to touch on some points here: - In Line 115, there is no clear definition of the \textbf{h-order heuristic} although it is mentioned between lines 26 and 31. If \textbf{h-order heuristic} is defined as, for example, "a heuristic requiring to know up to h-hop neighborhood of the target nodes", then do we really need Theorem 1? - In lines 274 and 275, it is stated that "We run all experiments for 5 times and report the average". It would be better if more runs are performed. - Lines 278-283: The training set size consists of the 90\% of all edges. I think it could be better to see the performance of the framework over different training/test set ratios. (In Appendix, there is another experiment similar to the one in Node2Vec paper in which 50\% of the edges are chosen as training set). - The standard deviation of experiments can also be included in the tables of the paper, as it is done in the Appendix. - The explicit features are not used in Table 2. I think it could be better to perform an experiment including explicit features. - The $\gamma$-heuristic theory is only used to show that small $h$ values can be sufficient to approximate some heuristics. It could be interesting that if the authors could develop a method that is able to learn the parameters within the general formulation. Clarity: I think that the structure and readability of the paper is quite good. The authors clearly define the \textbf{h-order heuristic} theory, with several examples. Then, they shortly mention about the methods that have been used in their framework. Here, I would like to indicate some mistakes in the notation: - In Lines 143 and 172, two different walk and length representations $|(w_1,...,w_n)|$ and $len()$ are used. - In Line 143, $d(i,x)$ and $d(i,y)$ should be $d(v_i,x)$ and $d(v_i,y)$ since $i$ is index. - In Line 143, the expression $l=|\{x,v_1,...,v_{i-1}\}| + |\{v_{i+1},...,v_{l-1},y\}|$ should be $l=|\{x,v_1,...,v_{i-1},v_{i}\}| + |\{v_{i}, v_{i+1},...,v_{l-1},y\}|$ - In Line 264, "This" should be changed with "These" Originality: The authors give a nice general formulation (\gamma-heuristic theory) for link prediction heuristics and show that many heuristics can be approximated with small $h$ values and appropriate parameters. Although it seems that the rest of the paper is similar to the WLNM method, they have managed to enhance it, outperforming previous state-of-the-art methods by changing the methodologies used at the different steps. Unlike WLNM, it also enables the use of explicit and latent features of nodes. Significance: As it was stated above, one of the major contributions of this work is the development of the $\gamma$-decaying heuristic theory which stems from the natural design of many heuristics. The proposed framework also allows us to benefit from the latent and explicit node features with the favor of GNN model in link prediction tasks and the experiments performed over many datasets also support its significance.

Reviewer 2



[Edit: the authors have nicely addressed my comments so I have upgraded my score] The problem of link prediction is a major graph mining problem. Recently, Zhang and Chen (KDD 2017) proposed a new framework for link prediction that leverages the subgraph of small diameter around the link to be predicted. This is achieved using graph neural networks, that get trained on various such subgraphs. In this work the authors extend the KDD paper in the following ways: first they provide a generic theorem (theorem 2) that essentially shows that for a general form of link prediction scores (equation 1), few hops around the link-to-be-predicted suffice. The authors proceed to show that several important existing link prediction methods fall under this general form, including Katz, PR, and SimRank. Then the authors propose the SEAL algorithm. While it uses neural networks, the way the data is fed to the NN is really neat. They provide an encoding of the subgraph network, as well as using any additional attribute information. The experimental results look nice, but as the authors discuss in the appendix, scaling to larger network is an issue for which they already propose certain heuristics. I enjoyed reading this paper, it is well written, and makes several clean contributions. On the negative side, it is incremental over the KDD 17 paper in my opinion, both conceptually and with respect to the experimental gains. Scalability is another issue too.

Reviewer 3



This paper studies the problem of link prediction, where instead of defining heuristic measures for such purpose, the authors propose a new framework to learn “heuristics” from local subgraphs. They justify their approach both theoretically and experimentally. I am reasonably familiar with the literature due to my own research in this area. Overall this is a very good paper. I particularly like the elegant theory that unifies a number of heuristic measures used for link prediction, which not only solidifies previous understanding but also inspires new thoughts. The SEAL framework proposed on top of the theory is a nice way of learning heuristics by combining both classic node-level features and a recent graph convolutional network approach. The experiments involve extensive comparisons with state-of-the-art and look convincing. Finally, the way the paper is presented, which uses concise language to convey meaningful insights, is much appreciated and makes the paper a joy to read. I therefore strongly recommend the paper to be accepted. Below are a few detailed comments that may further improve the quality of the paper: - lines 166 and 173. It is better to choose using either $|Gamma(j)$ or $|Gamma(v_j)$, but consistently; - line 254: it is not very clear how $X$ is constructed from what encoding vectors; - line 265: this trick is interesting and perhaps deserves further investigation. My intuition is that although it partially resolve the problem, it also renders the embedding features less meaningful; - in the experiments it would have been nice to see some “real world” link prediction task, where we aim for predicting links added in future snapshots of the network; - it would also be interesting to extend the current framework to the problem of predicting link removals, which together with predicting link additions may constitute a nice framework for predicting network evolution in general.